# OpenReview forum: "Psychometric Personality Shaping Modulates Capabilities and Safety in Language Models"
_ICLR.cc/2026/Conference — ICLR 2026 Conference Withdrawn Submission_

### Official Review · Reviewer_rSD5 · 2025-10-26

**Soundness:** 3
**Presentation:** 2
**Contribution:** 3
**Rating:** 4
**Confidence:** 3

**Summary:**

This paper investigates how prompting LLMs to adopt Big Five personality traits affects their performance on capability benchmarks (MMLU) and safety benchmarks (TruthfulQA, WMDP, ETHICS, Sycophancy).
The authors extend prior work on synthetic personalities in LLMs by examining downstream effects on safety-relevant behaviors, using validated trait markers and psychometric instruments like IPIP-NEO and SD3.
Experiments on models such as GPT-4 and the Llama family of variants show that personality prompts can decouple safety from capabilities: for example, low Conscientiousness drops safety and capabilities, while dark triad composites degrade safety without harming general knowledge.
Contributions of this paper include evidence challenging some of the safetywashing claims, implications for post-deployment steering, and calls for personality-sensitive evaluations.

**Strengths:**

1. The experimental setup is comprehensive, and the analysis is interesting and insightful.
2. It bridges the gap between established human psychometric theory (the Big Five) and the empirical study of LLM alignment, making it a creative and promising interdisciplinary contribution.

**Weaknesses:**

1. This paper’s approach is entirely based on prompt engineering. However, LLMs are very sensitive to prompts. This paper lacks a case study to analyze whether LLMs effectively answer questions or simply output irrelevant responses based on specific prompts, resulting in reduced accuracy (especially in models with smaller parameters, such as Llama-3-8B)
2. As a reader, I am more concerned with the Llama-3-8B model, which has relatively small parameters, but it is rarely analyzed in the main text.

Minor comments:
- Line 182: Typo: quantifiers -> qualifier, and from the example, the red part corresponds to the marker rather than the qualifier
- Line 320: Is it missing a reference?

**Questions:**

Please see Weaknesses above, add case studies, and analyze the specific reasons for the changes in performance on these benchmarks. Is the model providing valid answers, or simply refusing to answer, giving a perfunctory answer ("Who cares!")?

---

### Official Review · Reviewer_3JpW · 2025-10-30

**Soundness:** 1
**Presentation:** 2
**Contribution:** 1
**Rating:** 2
**Confidence:** 5

**Summary:**

This paper investigates how prompting large language models with personality trait descriptors based on the Big Five framework affects their performance on capability and safety benchmarks. The authors use Goldberg's trait markers (104 adjectives) to condition models with specific personality profiles (e.g., high/low conscientiousness, extraversion) and evaluate five models across benchmarks, including MMLU, TruthfulQA, WMDP, ETHICS, and Sycophancy.

**Strengths:**

1. The study addresses an interesting and important question: the relationship between language model personality and safety behaviors. This topic bridges psychological theory and AI alignment, making it both timely and significant for the community.

2. The use of established psychometric instruments such as IPIP-NEO and SD3 to assess and validate personality manipulations adds methodological rigor and helps ground the work in over a century of psychological research. This integration of validated psychological tools into AI evaluation frameworks is a valuable and commendable contribution.

**Weaknesses:**

1. The core contribution is essentially the application of established personality psychology prompts to existing benchmarks and the observation of correlations. However, the paper does not discuss several highly related prior works, including prompt-based approaches [1] and latent representation–based methods [2][3], which have already explored similar directions of controlling or inferring model personality.

2. The authors do not evaluate whether personality control through prompts is reliable or consistent, nor do they compare it with alternative personality control mechanisms (e.g., fine-tuning or latent manipulation). This omission limits both the novelty and credibility of the conclusions.

3. The experimental analysis lacks depth. The paper provides no explanation or theoretical reasoning for why personality prompts yield the observed safety or behavioral effects, leaving the mechanism largely speculative.

4. The result presentation is difficult to follow. Only Figure 2 summarizes the findings, yet the caption states that “columns are model families,” while the figure itself mixes individual model names and families. This inconsistency makes interpretation unclear and hinders reproducibility.

[1] Li, Guohao, et al. "Camel: Communicative agents for" mind" exploration of large language model society." Advances in Neural Information Processing Systems 36 (2023): 51991-52008.
[2] Chen, Runjin, et al. "Persona vectors: Monitoring and controlling character traits in language models." arXiv preprint arXiv:2507.21509 (2025).
[3] Ghandeharioun, Asma, et al. "Who's asking? User personas and the mechanics of latent misalignment." Advances in Neural Information Processing Systems 37 (2024): 125967-126003.

**Questions:**

Please check the weaknesses section.

---

### Official Review · Reviewer_ycgg · 2025-10-30

**Soundness:** 2
**Presentation:** 2
**Contribution:** 2
**Rating:** 2
**Confidence:** 3

**Summary:**

This paper examines how prompt-based psychometric personality shaping affects both the capabilities and safety performance of large language models (LLMs). Using Big-Five trait prompts (Openness, Conscientiousness, Extraversion, Agreeableness, Neuroticism), the authors evaluate multiple models—including GPT-4.1, Llama-3/4, and DeepSeek-V3—on MMLU, TruthfulQA, ETHICS, WMDP, and Sycophancy benchmarks. They report that decreasing Conscientiousness or Agreeableness sharply reduces safety scores (−20–40 pp) and that these changes occur largely independent of model scale or general capability. The paper concludes that personality shaping provides an orthogonal axis of behavioral control and calls for personality-sensitive safety evaluations.

**Strengths:**

The paper offers an intellectually engaging exploration of psychometric personality shaping in large language models (LLMs), connecting established psychological constructs (Big Five traits) to AI alignment and safety evaluation. The writing is clear, well-structured, and grounded in relevant theory, with polished figures and a strong interdisciplinary framing. The empirical findings, that reduced Conscientiousness or Agreeableness correlates with degraded ethical and truthful behavior, are intuitively compelling and align with known psychological evidence. The study also makes a valuable ethical contribution by discussing potential misuse (e.g., inducing adversarial personas) and proposing mitigations. While conceptually similar ideas exist, the authors effectively articulate why personality-sensitive evaluation may complement existing safety protocols, highlighting an underexplored but practically relevant perspective.

**Weaknesses:**

The paper’s central limitation is lack of novelty and empirical rigor relative to recent work. Several 2025 studies, most notably Li et al. (2025, BIG5-CHAT) and Handa et al. (2025, Personality as a Probe for LLM Evaluation), already demonstrate similar findings with stronger methodologies, including training-based or mechanistic personality shaping. By contrast, this submission relies solely on prompt-based manipulation, without robustness checks, significance testing, or multi-run validation. The personality validation via self-report questionnaires (IPIP-NEO, SD3) is conceptually interesting but circular, since models are prompted to reproduce those constructs. The study lacks control for prompt brittleness and does not distinguish stylistic mimicry from genuine behavioral change. Finally, claims such as “putting all safety benchmarks into question” are overstated relative to the descriptive evidence, which undermines the overall scientific precision of the contribution.

**Questions:**

1. How robust are the reported effects to prompt paraphrasing, random seed changes, or small lexical variations in the personality descriptions?
2. Can the authors provide statistical confidence intervals or variance estimates to support claims of “systematic” effects?
3. How does prompt-based shaping compare quantitatively with training-based or mechanistic methods such as those in BIG5-CHAT or Handa et al. (2025)?
4. Could the observed benchmark differences arise from changes in linguistic tone or verbosity rather than latent personality expression?
5. Have the authors examined whether these effects persist over multi-turn conversations or decay over time, and if so, how stable are shaped personalities?

---

### Official Review · Reviewer_rDDX · 2025-10-31

**Soundness:** 2
**Presentation:** 2
**Contribution:** 2
**Rating:** 4
**Confidence:** 3

**Summary:**

This paper investigates how psychometric personality shaping, based on the Big Five (OCEAN) framework, influences both capability and safety behaviors in LLMs. The authors design prompt-based personality modulation to simulate different personality traits and evaluate multiple models across a suite of benchmarks, including MMLU, ETHICS, TruthfulQA, WMDP, and Sycophancy.

The paper argues that personality shaping provides a lightweight, prompt-based mechanism for both behavioral steering and adversarial exploitation, motivating “psychometric control” as a new research frontier for LLM safety.

**Strengths:**

1. Psychological safety of LLMs is a timely and critical topic for AI safety.

2. This paper provides empirical results and analysis through psychological theory for LLMs.

**Weaknesses:**

1. Several related works are overlooked (e.g., Evaluating Psychological Safety of Large Language Models [EMNLP 2024]).

2. Analyses remain descriptive (percentage-point differences, heatmaps). Missing significance testing, confidence intervals, or effect-size analyses beyond normalized deltas.

3. Prompt-based methods are often not able to consistently maintain the traits of LLMs, and do not shift the internal property of LLMs. I would suggest the authors to further investigate based on the parameter-level analysis and try tuning-based methods.

**Questions:**

See above.

---

### Note · Authors · 2025-12-06

**Comment:**

It seems that the reviewers did not fully understand the intent of this study, which is partly our fault for not communicating it more clearly. We believe the paper fills an important niche that would be of interest to the community, but ICLR may not be the most appropriate venue for this particular formulation. We will revise the paper and submit it to a more suitable venue at a later time.

**Withdrawal Confirmation:**

I have read and agree with the venue's withdrawal policy on behalf of myself and my co-authors.